# Gender-Balanced Seats, Equal Power and Greater Gender Equality? Zooming into the Boardroom of Companies Bound by the Portuguese Gender Quota Law

**Sara Falcão Casaca** [1,*], **Susana Ramalho Marques** [1], **Maria João Guedes** [2,*] and **Cathrine Seierstad** [3,4]

1 Research Centre in Economic and Organisational Sociology/CSG–Research in Social Sciences and Management, Lisbon School of Economics & Management (ISEG), SOCIUS, Universidade de Lisboa, 1249-078 Lisbon, Portugal
2 Centre for Advanced Research in Management/CSG–Research in Social Sciences and Management, Lisbon School of Economics & Management (ISEG), ADVANCE, Universidade de Lisboa, 1249-078 Lisbon, Portugal
3 School of Business, Department of Business, History and Social Sciences, University of South-Eastern Norway, 3184 Borre, Norway
4 School of Communication, Leadership and Marketing, Kristiania College, 0107 Oslo, Norway
* Correspondence: sarafc@iseg.ulisboa.pt (S.F.C.); mjguedes@iseg.ulisboa.pt (M.J.G.)

**Abstract:** This paper seeks to analyse the potential for change in the gender quota law on corporate boards in Portugal. This is achieved by incorporating concepts and insights drawn from political science and the study of quotas in politics and adjusting these to the boardroom context. It adds to the literature on women on boards by shedding light on the importance of looking at descriptive representation, substantive representation, substantive equality and transformative institutional change, in order to understand a quota law's potential for eliciting gender balance in the boardroom, as well as greater gender equality in directorship positions, in board dynamics and at the workplace level. This study uses multi-strategy research methods. Evidence provided by the quantitative analysis of survey data, combined with the qualitative analysis of interviews undertaken with female and male board members and the contents of Gender Equality Action Plans (GEAPs), shows that there have been some changes in terms of descriptive representation, but fewer in relation to substantive equality, as men are still largely over-represented in positions associated with effective power and influence over decision-making. Moreover, although the promotion of gender equality at the workplace is valued by both groups, and particularly so by women, weaknesses have been found in the materialisation of such a commitment (substantive representation) through the adoption of GEAPs designed to tackle gendered patterns at the workplace (transformative institutional change).

**Keywords:** gender; women on boards; descriptive representation; substantive representation; substantive equality; transformative institutional change; gender mainstreaming; quota law

## 1. Introduction

In 2017, Portugal became one of the latest countries to introduce a law that established a gender quota on corporate boards, aimed at attaining—from 1 January 2018 onwards—a more balanced representation of women and men on the boards of directors and the supervisory bodies of public sector companies (amounting to at least 33% of the under-represented sex in state-owned and local government companies) and public listed companies (PLCs) (at least 20% and 33% of the under-represented sex, in 2018 and 2020, respectively). It also established the obligation for companies to design, implement and monitor Gender Equality Action Plans (GEAPs), as well as to make these publicly available on their websites. This additional legal obligation is unique when compared to other European gender quota laws (see Mensi-Klarbach and Seierstad 2020, for an overview), as it also seeks to encourage transformative institutional change in legally bound companies. Considering this new

legal framework as a "critical act" (Dahlerup 1988), this paper adds new empirical and theoretical insights to the women on boards (WoB) literature by exploring the links between greater gender balance on boards, substantive equality, substantive representation, gender mainstreaming in organisations and transformative institutional change.

Drawing on the contributions from gender studies, mainly in the domains of political science, organisations and institutional change, the paper is guided by the following research questions:

(1) To what extent is the gender quota law bringing about a re-composition of corporate boards, leading towards a greater gender balance (descriptive representation) and, at the same time, eliciting an equal share of positions of power and influence between male and female board members (substantive gender equality) in internal boardroom dynamics)?

(2) In what ways is an improved gender balance on boards producing greater interest among female board members and their male counterparts in the concerns of female workers and advancing gender equality at the workplace level (substantive representation)?

(3) How are the legally bound companies designing and adopting GEAPs as key strategic tools of a gender mainstreaming approach aimed at advancing gender equality at the workplace level (transformative institutional change)?

In addressing these questions, the paper goes beyond the already existing literature on the topic of the use of both gender quotas in general and gender quotas on boards in particular. This is achieved by expanding the relevance of the core conceptual tools beyond the context of politics, while also adding the potential for transformative institutional change, elicited by legally binding measures designed to accelerate the move towards a greater balance on corporate boards and gender-equal workplaces. Moreover, in addition to the core theoretical contributions on this topic, the paper incorporates the relevant political concepts found in key international political references in terms of gender equality and women's rights, such as the Convention on the Elimination of All Forms of Discrimination against Women (CEDAW), adopted in 1979 by the UN General Assembly, and the Beijing Declaration and Platform for Action, adopted by 189 UN Member States in 1995. The two conceptual legacies embedded in our analytical framework are the concept of substantive equality, from the first core policy document, and gender mainstreaming, inherited from the latter and applied, in this case, to the context of institutional change processes aimed at promoting gender-equal organisations.

With this paper, we demonstrate the importance of simultaneously focusing on descriptive representation, substantive representation, substantive equality and transformative institutional change. When these factors are taken together, we argue that it is possible to capture the real transformative potential (or failure) of a quota law designed to increase gender equality in the corporate boardroom and at the workplace. Besides adding to the literature on quotas on boards and women on boards (WoB), our findings may also help to inform policymakers and practitioners about how to align policy tools and actions in order to effectively elicit more equitable boardrooms and corporations in general.

## 2. Theoretical Framework

Since Norway's introduction of gender quotas for corporate boards in 2003, implemented in 2006 with a two-year grace period, the political discussion of how and why to increase the share of women in the upper echelons of the company hierarchy has flourished, both at a national, as well as a transnational, level. In 2012, Vice-President Viviane Reding, the EU's Justice Commissioner, suggested a proposal for quotas—the so-called Women on Boards Directive proposal (European Commission 2012).

While such a proposal did not originally receive sufficient support from the member states, there have, nonetheless, been several contentious debates about introducing quotas at an EU level, as neither the share of women on boards nor gender equality in organisations have improved considerably. In the European Union, seven countries have adopted what can be understood as *hard* regulatory frameworks (mandatory quotas, backed by

some sanctions for non-compliance). These countries are Belgium, Italy, France, Germany, Austria, Portugal and Greece. Spain and the Netherlands, together with the EEA country Iceland (2010), have introduced *soft* quotas (mandatory quotas that are not backed by sanctions) (EIGE 2020). Moreover, some countries (e.g., the UK) have introduced voluntary targets with the aim of increasing the share of WoB in the largest companies. Consequently, different types of policies have been introduced in European countries over the last 20 years.

In June 2022, the directive on improving the gender balance among non-executive directors of listed companies, proposed by the Commission in 2012, was finally agreed upon and quotas are now to be implemented at the EU level. This means that the push for further gender equality is firmly on the agenda in Europe and there is a need for greater knowledge about different countries' experiences with the use of quotas.

The spread of quotas at the national level has been widely explored in the literature, especially in the context of politics (Childs and Krook 2006; Krook and Zetterberg 2014a, 2014b), and, over the last two decades, also in the context of corporate boards (see, for example, Kirsch 2018, for an overview). One early branch of research that focused on gender quotas on corporate boards and/or women on boards looked at the effect of women on boards and the differences between male and female directors following the introduction of quotas. In particular, a wide range of studies have sought to analyse the impact on corporate economic performance of a greater gender balance in the boardroom (e.g., Ahern and Dittmar 2012; Bøhren and Staubo 2016; Magnanelli and Pirolo 2021; and Ferreira 2015, for a critique). Another branch of research has sought to look beyond the numbers, focusing on the diverse institutional frameworks in which quotas have been introduced and operate (Humbert et al. 2019; Mensi-Klarbach and Seierstad 2020; Terjesen et al. 2015) and the role of actors pushing for change (Doldor et al. 2016; Seierstad et al. 2017). What seems to be largely missing in the context of quotas on boards/WoB research is a focus on the legislative effects on a variety of representative processes, as found in the political science literature. In this domain, research has been focusing on the impact of binding affirmative action measures (quotas) on representation (especially in the political spectrum), whether this is descriptive, substantive or symbolic in nature (Childs and Krook 2006; Krook and Zetterberg 2014a, 2014b). Building on these ideas, this paper seeks to explore the relevance of such conceptual tools[1], by also incorporating three additional concepts—substantive equality, gender mainstreaming and transformative institutional change—to the analysis of the effects generated by the implementation of the gender quota law aimed at corporate boards in Portugal.

Our analytical framework was built upon the following concepts and theoretical insights. Descriptive representation refers to the increasing number of women serving on boards, as a result of binding legal measures, such as the gender quota law in place in the country; the term is used to describe the changes occurring in the gender composition of boards, including men and women's characteristics in terms of age, qualification and boardroom experience (Krook and Zetterberg 2014a). Substantive representation relates to whether women's issues at the workplace are considered to be priorities of these newly appointed women and whether they are taken into account in the decision-making process (Krook and Zetterberg 2014a). In this regard, the concept of "critical mass" is also fundamental, as it has been used to explore whether women in positions of power do, in fact, represent the interests of other women and seek to produce outcomes that will address their concerns (Kanter 1977). It presumes that women are more likely to speak out on behalf of other women when they represent not only a few "tokens", but actually amount to a considerable number of women—i.e., forming at least part of a "tilted group" (35–65%). In these groups, women are more likely to express their individuality and contribute to the group dynamics. On the contrary, when women are perceived as mere tokens, their actions are more visible. Under these circumstances, women may feel the pressure of being under scrutiny, which may lead them to shy away from raising their voices and expressing their ideas (Kanter 1977; Konrad et al. 2008; Santos and Amâncio 2014). Adapting and extending Kanter's theory to the political domain, Dahlerup (1988)

posits that 30% is the critical threshold for determining women's influence on the political agenda. In such circumstances, they would be able to join forces, build alliances and work more effectively together to advance gender equality and women-friendly policies, also being able to influence their male colleagues to accept and approve such policy changes. However, this same author states that other society-level and institutional factors, besides the actual numbers, might explain changes in policy or indeed the lack of change, emphasising the role of critical acts and initiatives (see also Childs and Krook 2008). Other scholars have, however, underlined the limited impact of numbers and stressed the importance of broader factors of opportunity and constraint, such as already existing organisational structures and cultural norms (Acker 1990; Correll 2017), as well as exclusionary practices (Bendl and Schmidt 2010).

Drawing on *The Convention on the Elimination of All Forms of Discrimination against Women* (CEDAW (UN 1979)), adopted in 1979 by the UN General Assembly, we view substantive change as the change that produces "women's de facto equality with men", hence, substantive equality. Once transposed to our research focus, substantive change towards gender equality would occur when male and female board members have access to equal board positions and exert an equal influence in terms of setting the board agenda and determining the decision-making processes and results. Transformative institutional change is a concept that is embedded in organisational studies and is based on the theoretical assumption that organisations are not gender-neutral, but are instead gendered institutional settings where gendering processes generate and reproduce gender inequalities (Acker 1990), "gender regimes" (Connell 2002), "inequality regimes" (Acker 2006) or "gender factories" (Calás et al. 2014). Barriers inside organisations are not inert and static, but fluid and dynamic (Bendl and Schmidt 2010). Exclusion and discrimination are processes that can be questioned, transformed, or even eradicated. Revising institutional policies, practices and work routines is a fundamental incremental process, designed to overcome hegemonic gender beliefs and male-dominant organisational structures and cultures (Ely and Meyerson 2000), undo their gendered patterns (Britton 2000; West and Zimmerman 1987) and create gender-equal organisations (Correll 2017). An important element in this regard is the concept of gender mainstreaming—a legacy of the Beijing Declaration and Platform for Action adopted by 189 UN Member States in 1995; more recently applied to institutional settings (EIGE 2020; European Commission 2016; ILO 2012). It is a systematic and coherently articulated gender equality perspective adopted by an organisation, at all levels and in all areas of activity. Its adoption requires the following: (a) a full and comprehensive diagnosis seeking to uncover the internal factors that (either directly or indirectly) constrain equal opportunities, equal treatment and equal results between men and women; (b) planning, reorganisation, change and improvements in terms of decision-making, policies, practices and processes, in order to ensure the full and systematic integration of a gender equality perspective at all levels and in all organisational areas; (c) a systematic monitoring and evaluation process. Therefore, in this regard, the design and adoption of GEAPs, grounded in the evidence provided by organisational diagnosis—ideally undertaken in the form of a gender audit (ILO 2012)—is a key tool in an institutional gender mainstreaming approach, designed to promote gender-equal organisations (Casaca and Lortie 2017).

There is an important body of research that demonstrates that a greater representation of women on corporate boards has the potential to improve the situation of women below the boardroom level. This might narrow the gender pay gap (Cohen and Huffman 2007), increase the number of women in other managerial positions at the workplace and benefit all other non-managerial female workers (Skaggs et al. 2012). Research has, therefore, shown that women on boards tend to act more as "agents of change" than as "cogs in the machine", thereby actively helping to tackle gender segregation at the workplace and to challenge the gendered organisation (Stainback et al. 2016). Nevertheless, the effects of women's representation on boards in terms of producing greater equality in organisations have also been questioned (e.g., Bertrand et al. 2019; Seierstad et al. 2021), indicating that

the focus on boards needs to be complemented with further organisational initiatives and policies. This makes the Portuguese case particularly interesting, together with its introduction of a gender quota law that also makes the adoption of GEAPs mandatory.

### 3. Setting the Context: The Gender Quota Law in Portugal

Although the implementation of a gender quota law on corporate boards is a recent policy measure in Portugal (Casaca 2017; Espírito-Santo 2018), previous normative instruments that support self-regulation have been in place for quite some time. In 2013, the obligation for state-owned companies to design, implement and monitor GEAPs and to ensure a plural presence of men and women on boards was legally established in the country for the first time (Decree-Law No. 133 2013), following various recommendations on GEAPs issued by Government Resolutions in 2007. In 2012, a Government Resolution (RCM No. 19 2012) was the first normative instrument in the country to mention publicly listed companies (PLCs), explicitly recommending the adoption of GEAPs and self-regulatory measures to achieve a plural presence of women and men in management and supervisory bodies (no specific target was specified). Accordingly, three years later, in 2015, the Government took the first steps with PLCs towards reaching a commitment that would promote a greater balance in the representation of women and men on their boards of directors/management bodies (RCM No. 11-A 2015, of 6 March). On the part of the companies, this commitment meant that they would be bound to a target representation of 30% of the under-represented sex, by the end of 2018. However, these first steps did not have the desired effects, as only 13 out of 39 PLCs responded positively to the appeal to sign this commitment.

As the incentives to self-regulation did not produce the desired results, in 2017, Portugal became one of the most recent countries to introduce gender quotas on corporate boards, with the passing of Law No. 62 (2017), on 1 August, which established gender quotas for attaining a balanced representation between women and men on the boards of directors and the supervisory bodies of the public sector (state-owned and local government) companies and PLCs. For state-owned and local government companies, the minimum proportion of men and women appointed to such bodies should not be lower than 33.3%, as from the first elective general meeting held after 1 January 2018. The law applies to new appointments (or reappointments) and not to mandates in progress.

In the case of PLCs, the implementation of the law was divided into the following two stages: in stage one, after the first elective general meeting held after 1 January 2018, the proportion of members of each sex appointed to each management and supervisory body should not be lower than 20%; in stage two, after the first elective general meeting held after 1 January 2020, this proportion should not be lower than 33.3%. These targets are tilted (Mensi-Klarbach and Seierstad 2020), being well below the minimum parity threshold of 40% (Council of Europe 2003), which is also the quantitative objective established in the 2012 proposal for the EU Women on Boards (WoB) Directive (European Commission 2012).

The law provides for the application of both reputational and financial sanctions in the event of non-compliance, but only in relation to the minimum thresholds of representation; no sanction is imposed for the non-implementation/publication of GEAPs. Non-compliance by state-owned companies results in the invalidity of the board's nomination and a 90-day period for a new nomination. In the case of PLCs, the Portuguese Securities Market Commission (CMVM) first issues a declaration reporting non-compliance, prior to establishing a 90-day period during which the board's nomination must be changed. Continued non-compliance determines the application of a publicly registered reprimand to the offender. In the event of non-compliance by a PLC for a period exceeding 360 days from the date of the reprimand, the Securities Market Commission applies a compulsory pecuniary penalty, not exceeding the total of one month of remuneration of the respective management or supervisory body, for each semester of non-compliance. The coverage of the normative framework is low, as the law applies to less than 0.05% of the whole business universe in the country.

Additionally, the new normative framework established the obligation for companies to design, implement and monitor annual GEAPs, and publish them on their website. This legal obligation is quite unique when compared to other legal frameworks of gender quota laws, as it seeks to promote transformative change in organisations (EIGE 2020; European Commission 2016). As a result of Legislative Order No. 18 (2019), the Commission for Equality in Labour and Employment (CITE) made available an instrument specifically designed for legally bound companies, the "Guide for the Development of (annual) Gender Equality Action Plans". Once the internal diagnosis has been performed, companies must prepare the respective GEAPs, which must be made available on the company's website and communicated, on an annual basis, to the official bodies in charge of gender equality in the country.

## 4. Methodological Options

In order to explore the effects of the gender quota law in Portugal, a combination of quantitative and qualitative methods was adopted at different moments in the research process. In order to measure Portugal's progress in terms of WoB and the respective descriptive representation, including comparison with its European Union counterparts, data were retrieved from the Women on Boards-PT Database[2] and the Gender Statistics Database, provided by the European Institute for Gender Equality (EIGE) for the largest PLCs. Moreover, a complementary survey was designed and sent out to male and female board members. In-depth interviews were also undertaken with both male and female directors. Finally, GEAPs were collected from the companies' websites, whenever these were made publicly available, and their content thoroughly analysed (as detailed later on in this paper).

*Data Collection and Analysis*

As a starting point for the beginning of the empirical research, in the first quarter of 2019, all the legally bound companies—PLCs, state-owned and local government companies—were contacted by telephone in order to obtain the board secretariat's email address for the purposes of future communication. A document containing information about the project was then distributed via the newly obtained mailing list, as well as an invitation addressed to each company to fill in an online survey and to further participate in an individual interview. Following these contacts, the survey was distributed via an anonymous link to 230 members (W = 87 and M = 143) of the management and supervisory bodies of PLCs, state-owned and local government companies. A total of 161 valid responses were obtained (W = 84 and M = 77). Considering the total number of directors (excluding alternate directors) in the years in which the survey was applied, the survey respondents represent approximately 9% of the members of the universe of PLCs and 10% in the case of the public sector companies (state-owned and local government companies). The respondents were between 30 and 81 years of age (mean = 52 years) and, when broken down by sex, male respondents were older than women (average age of 54 and 50, respectively).

In addition, 43 semi-structured interviews were held between November 2019 and November 2020 with members of the boards of directors and supervisory bodies of companies covered by the Law (W = 22 and M = 21), aged between 33 and 69 years (mean = 52 years), with the average age of women and men being 50 (age range 38–61 years), and 54 (age range 33–69 years), respectively. The interviews were fully transcribed and kept anonymous by assigning a code to each person interviewed, so that a thematic analysis could then be undertaken. At this stage, a flexible approach was adopted with a view to combining both a deductive and an inductive approach; even though the research questions, guided by the theoretical framework, played an important role in the process of coding, the thematic analysis process was kept open to a data-driven approach (Braun and Clarke 2006). A series of themes (patterns) commonly expressed in the discourse of the interviewees were, therefore, identified and analysed and were as follows: (a) the perception of the power to

influence decision-making processes; (b) the perceived constraints on gender equality in intra-board dynamics; (c) a commitment towards gender equality beyond the boardroom and the most relevant subjects for intervention at the workplace le vel. The qualitative data analysis was computer-assisted (MAXQDA). Furthermore, a total of 101 GEAPs were collected from the companies' websites (28 from PLCs companies, 48 from state-owned companies and 25 from local government companies), in 2020, and their content analysed in terms of their implementation potential. For that, as detailed later in Section 5.4, we observed whether the outlined measures were specific, measurable, attainable, relevant and time-bound (EIGE 2016).

## 5. Results and Discussion

### 5.1. Descriptive Representation—Towards Greater Gender Balance on Corporate Boards

As mentioned earlier in this paper, prior to the introduction of the gender quota law in Portugal, policies to promote greater gender balance on the boards of publicly listed companies relied on recommendations and incentives for self-regulation. The practical results of these soft measures fell short of what was intended (Casaca 2017). In 2012, the year when these incentives were introduced, women accounted for only 7.4% of the board members of the largest PLCs; three years later, the numbers were about 6 percentage points (p.p.) higher, amounting to 13.5%, yet, even so, still almost 10 p.p. below the average EU figure for that year, 22.7% (Figure 1). Since then, progress has been quite notable in terms of the number of female board members in Portugal. In 2018, the year when the quota law came into force in the country, the country began to narrow the gap with the EU and, by 2021, the difference was approximately 1.9 p.p. below the European average for the same year.

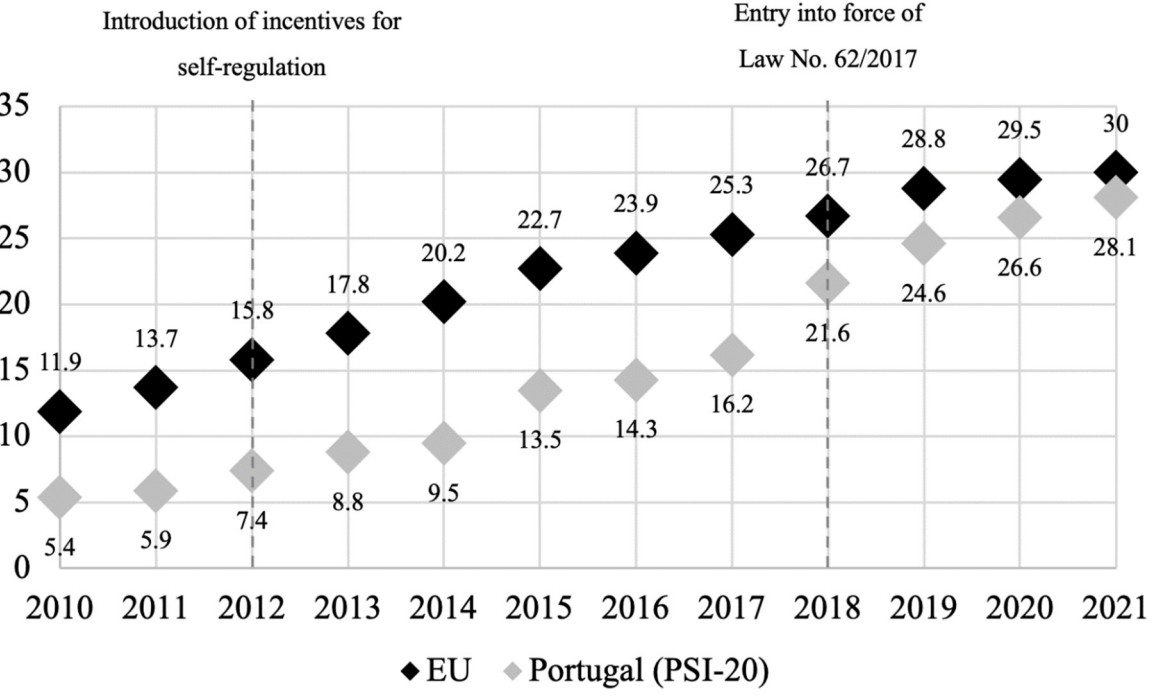

**Figure 1.** Percentage of women board members in the largest PLCs in Portugal and the European Union (2010–2021). Statistics for the EU in 2020 and 2021 refer to EU-27. Data for this figure were obtained from the European Institute for Gender Equality (EIGE 2021), *Gender Statistics Database*.

Focusing exclusively on the national landscape, in the case of the overall group of PLCs, when the law came into force in 2018, only 18% of board members and 19% of the members of corporate supervisory bodies were women (Appendix A, Table A1). In the particular case of companies that became legally bound by the normative framework in

that year (N = 14), women represented 22% of all board members, a figure that was already above the minimum threshold of 20%. Two years later, in 2020, the numbers rose to 26% of women on the boards of directors and 32% on the supervisory bodies. This was the transition year to the new target of 33.3%. Of the 38 listed companies, 33 were already bound by the law; 23 of these had to comply with the minimum threshold of 20%, while 10 were obliged to reach a minimum level of 33.3%. In the latter group, the percentage of WoB was below the target required by law for the boards of directors, with only 31% of women, whereas, in the supervisory bodies, the threshold was 33%.

In the state-owned companies, women represented 40% in 2020, which was above the minimum threshold of 33.3% required by law (Appendix A, Table A2). In the group of companies bound by the law (N = 99), women represented 44% of the members of the board of directors; in local government companies, women represented 29% of board members. It is not, however, possible to present data for the companies to which the law is applicable, since information on the starting dates of mandates was practically non-existent in the sources consulted. As far as descriptive representation is concerned, the data show that the law has prompted an increase in women in all groups of companies, with the management and supervisory boards of companies bound by the law in Portugal being slightly more gender balanced.

One notable aspect is that the increase in the number of women on boards is not the result of an increase in the board size. In a related analysis, Casaca et al. (2022) showed that women who were appointed before the law came into effect were more frequently found on larger-sized boards in comparison with men, and the differences were statistically significant. However, for those appointments made after the law came into effect, and although women were still to be found on larger-sized boards, this gap was shortened and became statistically non-significant.

*5.2. From a Greater Gender Balance to Substantive Equality?*

In this study, we seek to explore a multitude of changes as a result of the implementation of the gender quota law in the country. It is important to consider the potential effects of a legislative change, since a greater gender balance—expressed in the form of descriptive representation—does not necessarily result in greater gender equality (substantive equality)—i.e., equal positions of power and influence in the boardroom for men and women. In this regard, it should be noted that their entry into the management and supervisory bodies of the largest PLCs in the country has been, above all, to positions with non-executive functions. Hence, in 2018, women accounted for only 10% of executive positions, an increase of 0.4 p.p. since 2012, while, at the same time, representing 29.1% of non-executive board members. This latter figure represents a growth of more than 22 p.p. over the same timespan (Figure 2). In 2021, the percentage of women in executive positions at these companies fell to 14.5%, while in non-executive positions, there was an increase to 37.8%. In state-owned companies, the situation is somewhat different, as this was the only group where there were more women in executive than non-executive positions, albeit by only a small margin (Appendix A, Table A2).

A similar result is found for the whole group of PLCs, where women are being appointed to mainly non-executive positions, accounting, in 2018, for 27% of the total number of board members filling such positions, and for 36% in 2020. In 2018, women held only 9% of executive board positions, increasing to 14% in 2020. For this group of companies, there was only one female CEO and one female Chair for 2018, increasing to two in 2020 for the latter position (Appendix A, Table A1).

As previously mentioned, in the public sector in 2020, the group of state-owned companies was the only one where there were more women in executive positions (41%) than in non-executive ones (38%), albeit by a small margin (Appendix A, Table A2). As far as women serving as chairpersons are concerned, there was a total of 31 (in 173 companies), with a total of 41 serving as chairpersons of the supervisory body. In local government companies, 29% of executive positions were occupied by women. In non-executive roles,

women amounted to 37%. Additionally, a total of 25 women (in 181 companies) were chairpersons.

The data, therefore, show that progress has been slow, with there still remaining a remarkably large gender gap. Such a result clearly calls for an increase in the representation of women in positions where they have effective power and influence over decision-making (e.g., executive roles, CEO, chairperson). As we sought to go beyond the focus on descriptive representation on boards by capturing, understanding and comparing the perceptions and experiences of female and male board members, the quantitative analysis of the survey data was combined with the content analysis of interviews held with both groups of directors.

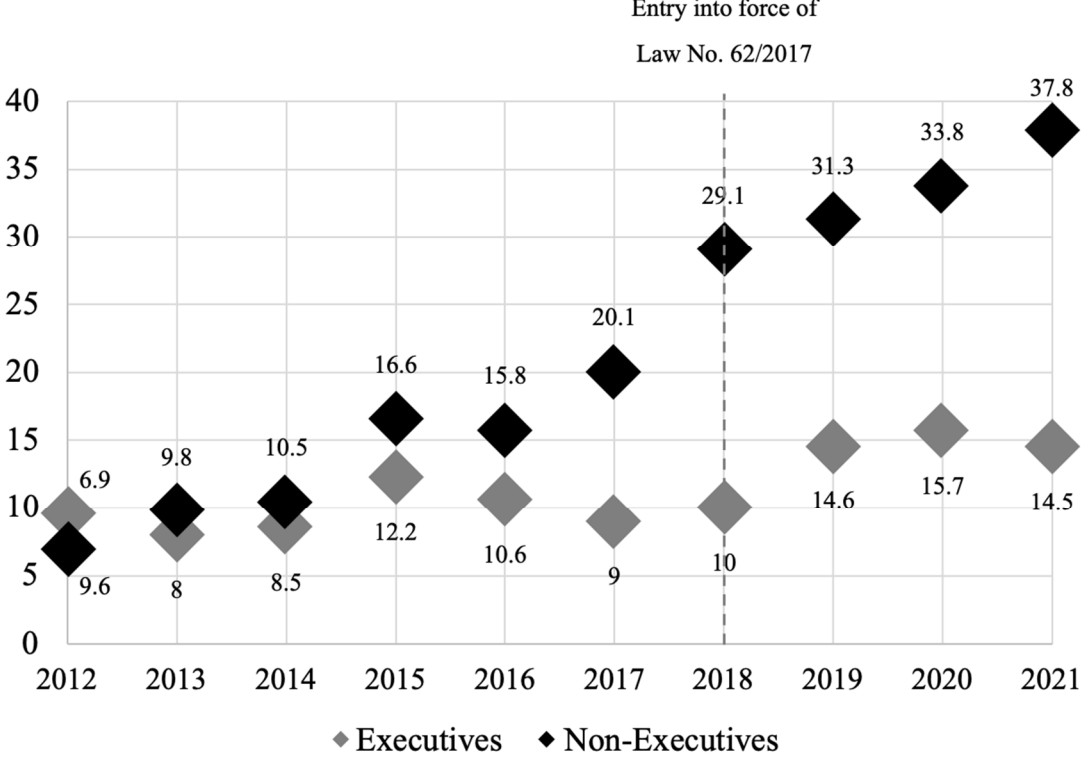

**Figure 2.** Percentage of women board members in the largest PLCs in Portugal in 2021, including executive and non-executive positions. The source does not provide information disaggregated by type of position for the period prior to 2012. Data for this figure were obtained from the European Institute for Gender Equality (EIGE 2021), *Gender Statistics Database*.

As Table A3 in Appendix B shows, the statements with which the survey respondents tended to agree the most were as follows: *I always share my opinion, regardless of whether or not it is the same as that of my colleagues, I am always given the opportunity to share my opinion* and *I feel that I am treated with respect*. Men agreed slightly more with the following statements: *I always share my opinion, regardless of whether or not it is the same as that of my colleagues* and *I am always given the opportunity to share my opinion*, while women agreed more with this latter sentence, and the statement *of I feel that I am treated with respect*. However, the mean equality test showed no statistically significant differences between female and male board members. Most survey respondents considered that women and men are equally active in the meetings of the body to which they belong (see also Table A4—Appendix B). These results, therefore, are in line with previous research findings, according to which both male and female board members perceive that they have a high level of information sharing and a high level of influence (e.g., Elstad and Ladegard 2012).

The qualitative data shed further light on people's experiences and provided stories of how these unfolded. As far as women are concerned, the following statements from interviews reinforce the evidence of a common perception:

> *Oh yes, clearly. I have a lot of influence. We try to make sure decisions are consensual, and when they are not consensual, they stay on hold, and, the next week, at the next board meeting, or even informally, at lunch, in a cafe, we debate them again, and we come back to work on them. But there are many occasions when the first decision is not consensual, and it is necessary to talk a little. Mainly for my colleague: he clearly preferred tighter portfolios, and the maintenance of some status quo from the past, and this implies a greater openness for discussing issues.—W_SOC13[3]*

> *Oh, completely (having an influence on decision-making). On top of that, I have a very strong temper, and, therefore, I find it very difficult not to argue to exhaustion something that I defend. And so, I also know how to respect collegial decision-making, and try to find a point of consensus that I feel comfortable with. But I feel that I have the possibility, the opportunity, to express my point of view, and I have respect from my colleagues, and in particular from the chair of the board of directors, who respects me a lot and listens to my opinions.—W_SOC17*

Interestingly, however, the statements also point to how, beyond such a scenario of apparently equally influential voices, gender inequalities might be reproduced behind the formal scenes, as women lack access to the traditional "old men's club" of networking, demonstrated by the following statements:

> *And then there's another thing, I think we women do less networking, and now, after all these years, what's the conclusion I've come to? Nothing important is discussed in meetings; important things are discussed either beforehand or afterwards. So, if you want to have an impact, if you want to change something, you have to lobby people one by one. If you want to say something important at the meeting and have someone listen to you, you have to talk to three or four people beforehand.—W_PLC15*

> *I would say, clearly at the executive committee level and then on the board, behind-the-scenes work is essential.—W_PLC30*

It seems, however, that numbers do matter. The women with board experience who were interviewed tended to highlight the negative effects of a "token experience", which is also in line with previous studies (e.g., Konrad et al. 2008), shown by the following statements:

> *If I was alone? It would be harder, much harder. I have no doubt that it is not indifferent whether you have more than one woman. By the way, my life was not very easy at the beginning, and if it weren't for my female colleague, who is extraordinary, and who supported me from the very first moment, things would have been a lot more complicated. And so, yes, it's true that it wouldn't be the same if I was alone.—W_SOC7*

> *My problem is that sometimes I feel very lonely because I am sometimes the only one in a meeting with the Board of Directors (. . . ) To be honest, I often feel that I am alone; in the sense that this doesn't seem very fair to me, and I'm not an ornament, I'm a thinking human being, who actually thinks quite well.—W_LGC10*

> *When I joined the company, it was all men and me. (. . . ) I spent two years alone; it was an otherworldly thing. Men used to make a few jokes from time to time, and then apologise. I felt that this wasn't my place. If they make these kinds of jokes, well, then that's pretty nasty.—W_PLC15*

*5.3. From Greater Gender Balance to Substantive Representation?*

On average, the survey respondents tended to attribute great importance to gender equality in the company, with women giving more importance to this question than men (see Appendix B, Table A5). There were no statistically significant differences in the way

that men and women perceived the importance given to gender equality by the different members of the corporate board (Table A6). Except for the importance given to gender equality by the chairperson/CEO, which had a similar classification for both women and men, all of the other cases were ranked slightly higher by women.

The qualitative data support the quantitative findings, and the interviewees in general highlighted the fact that commitment towards gender equality at the workplace is not a "women's issue", as shown by the following statements:

> *I would say that the concern with gender equality in the company is not only present in the minds of the female members of the board, but also in those of the male members. The President, in particular, is extremely sensitive and concerned about these aspects. So, it absolutely cuts across both genders.* —W_SOC5

> *This commitment does exist, and it is reflected in our decisions, too; decisions on hiring, developing activities, training, raising awareness and creating opportunities, so that there is no discrimination whatsoever in having a Gender Equality Action Plan. In fact, this commitment is in our own code of conduct.*—M_SOC42

The survey respondents tended, on average, to rate the issues relating to tackling the gender pay gap at the workplace as the most important ones, although all other matters raised were also considered to be generally important (Table A7). Furthermore, interviewees, in general, recognised the need for concrete measures leading to transformative institutional change, as demonstrated by the following statements:

> *I think the measures [of the Gender Equality Action Plan] that I value most were the ones that worked towards tackling the gender pay gap. Because our first temptation was to think there was no problem. And then the Sustainability Director herself was doing some research based on the data that are publicly known, working with the colleague from Human Resources, and we came to the conclusion that in fact there are still some issues that, from our point of view, need improvement.*—W_SOC4

> *There is another theme that also needs to be addressed: I can have gender diversity and a gender pay gap, as English law firms did, for example, where the partners earned between 32% and 42% more than other board members at the same hierarchical level. The board of directors has to be strongly committed and request the executive committee to take the necessary measures to close these gaps.*—M_SOC26

There are statistically significant differences to be noted between the answers provided by female and male board members ($p < 0.01$). Women attributed greater importance to the promotion of a balanced participation of women and men in top decision-making positions and also in management and leadership positions (Appendix B, Table A7).

*5.4. Is There Room for Transformative Institutional Change?*

The inclusion of the adoption of a GEAP as a requirement of the gender quota law was an important step, provided that it is not viewed as a document merely designed to comply with a formality or a legal imperative, but is regarded as a strategic instrument for mainstreaming gender in organisations and for eliciting transformative institutional change (EIGE 2016, 2019; ILO 2012). However, as of December 2020, the compliance with this obligation on the part of legally bound companies was low, particularly among public sector companies; 73.7% of PLCs, 27.7% state-owned companies and 13.8% of local government companies adopted GEAPs and made them publicly available on their websites.

In the content analysis, we checked whether the measures outlined in the GEAPs were specific, measurable, attainable, relevant and time-bound (SMART) (EIGE 2016), enabling the systematic monitoring and follow-up of the transformation process, or, instead, whether their content was vague and close to being little more than a mere "declaration of intent". Specifically, a measure was considered to have been formulated in such a way as to contribute to the effective promotion of equality between women and men when it was associated with the following six categories that enabled its monitoring: (i) departments

and people responsible; (ii) human and financial resources (if necessary); (iii) objectives; (iv) timing; (v) goals; (vi) result indicators.

In order to assess the implementation potential of the GEAPs, the following five criteria were defined:

- High implementation potential: GEAP in which the formulation of at least 90% of the measures is complemented with information on the six categories that enable their monitoring;
- Moderately high implementation potential: GEAP in which the formulation of two thirds of the measures (but fewer than 90% of all measures) is complemented with information for all the categories that enable their monitoring;
- Moderate implementation potential: GEAP in which the formulation of at least half of the measures (but fewer than 66% of the total) is complemented with information for all the categories that enable their monitoring.
- Moderately low implementation potential: GEAP in which the formulation of at least one third of the measures (but fewer than half of all measures) is complemented with information for all the categories that enable their monitoring.
- Low or zero implementation potential: GEAP in which the formulation of fewer than one third of the measures is complemented with information for all the categories that enable their monitoring.

In half of the 28 PLCs that published their GEAPs, these documents had high (4 companies) or moderately high (10 companies) implementation potential. There is, therefore, room for improvement in strengthening their potential to bring about transformative institutional change. In the case of state-owned companies, in half of those that disclosed their GEAPs (24 out of 48 GEAPs), there was high (15 companies) or moderately high (nine companies) implementation potential. There are, however, weaknesses that need to be overcome, since, in 12 companies, the potential of their GEAP is moderately low or low/zero, and in the remaining ones (also a quarter of all the companies under analysis), this potential is only moderate. In local government companies, more than half of the GEAPs have moderately low or low implementation potential (out of 25 GEAPs). Only two companies have GEAPs with high implementation potential. On the other hand, the GEAPs of 13 companies have moderately low or practically zero effectiveness potential. The remaining companies have GEAPs with moderate (four) and moderately high (six) effectiveness potential.

The low number of GEAPs disclosed by legally bound companies, especially in state-owned and local government companies, suggests that there is still much to be achieved in terms of raising awareness and training critical actors. Furthermore, especially in this universe of companies, there are weaknesses in the potential of the measures contained in the GEAPs for bringing about transformational institutional change.

## 6. Concluding Remarks

We have adopted and amended concepts and ideas from political science and the use of quotas in politics (e.g., Childs and Krook 2006; Krook and Zetterberg 2014a, 2014b) to the context of gender quotas on corporate boards. We believe that this provides valuable nuances for better understanding changes and the effects of gender quotas on boards, beyond numerical changes, financial results or simplistic ideas of equality, which have dominated much of the WoB research undertaken to date. In particular, our study demonstrates the importance of looking at change in terms of descriptive representation, substantive representation, substantive equality and transformative institutional change. Taken together, these aspects capture the real transformative potential (or failure) of a quota law designed to increase gender equality in the corporate boardroom and at the workplace.

The gender quota law is stimulating a re-composition of corporate boards towards greater gender balance in Portugal, so that more women are now represented in the highest positions of power. Change has, therefore, taken place in terms of descriptive representation. At this first stage in the implementation of the gender quota law, progress has,

however, been rather slow in terms of executive positions, CEOs and chairpersonships. There still remains a remarkably large gender gap (which is similar to what has been found in other countries following the introduction of a quota—see Seierstad et al. 2021). Most newly appointed women have filled either supervisory or non-executive managerial positions. Consequently, the law has been less successful in bringing substantive equality to the boardroom, as only a small number of women are actually filling positions that allow for the exercise of effective power and influence over decision-making. Despite this finding, among those serving on the management body, both male and female board members recognise that they have a high level of information sharing and a high level of influence. On the flipside, however, they also acknowledge—particularly women—that most of the key strategic information and decision-making processes are conducted behind the formal scenes of the boardroom. Such a common statement suggests that in-group preference dynamics (Kanter 1977) may disadvantage women in terms of access to strategic information and power, thereby reproducing exclusionary practices (Bendl and Schmidt 2010).

According to both men and women's narratives—slightly more so in the case of women—gender equality at the workplace is particularly highly valued. In this regard, tackling the gender pay gap at the workplace is viewed as the most important aspect of gender inequality in need of intervention. The practical effects of such a strong attitudinal commitment towards gender equality and female workers' concerns (substantive representation) have so far been largely timid. The adoption of a GEAP is a requirement of the gender quota law, viewed as a strategic instrument for mainstreaming gender equality in organisational policies, practices and processes. However, compliance with this obligation on the part of legally bound companies has been low, particularly among state-owned companies and local government companies—paradoxically, the very segments that should be exemplary in their gender mainstreaming approach. Our findings suggest that there is ample scope for improving the transformative institutional change potential of such companies, signalling that there is still much to be changed in terms of an effective board members' commitment to undoing the gendered patterns that exist at the workplace level. As the gender quota law is relatively recent in the country, changes at the three levels (descriptive, substantive and transformative) may still be only gaining very initial momentum. Further research should be designed to track future gender dynamics both in the boardroom and at the workplace level. The current findings may help to inform policymakers and practitioners about how to adjust and align current policy tools and actions in order to effectively elicit more equitable boardrooms and corporations in general. One policy adjustment should be to revise the mandatory quota law so that that compliance in the adoption of GEAPs will be effectively ensured. Moreover, more practical support could be given by official public bodies in charge of promoting gender equality through the provision of training activities to companies, outlining the objectives of a gender mainstreaming approach in organisations and showing how to design, implement and monitor such plans in order to bring about gender-equitable workplaces in Portugal.

**Author Contributions:** Conceptualization, S.F.C. and C.S.; methodology, S.F.C., S.R.M. and M.J.G.; software, S.R.M.; validation, S.F.C., S.R.M. and M.J.G.; formal analysis, S.F.C., S.R.M. and M.J.G.; investigation, S.F.C., S.R.M. and M.J.G.; writing—original draft preparation, S.F.C., S.R.M.; writing—review and editing, All authors; supervision, S.F.C. and M.J.G.; project administration, S.F.C. and M.J.G.; All authors have read and agreed to the published version of the manuscript.

**Funding:** The authors acknowledge the valuable research funding support provided by FCT, I.P., the Portuguese national funding agency for science, research and technology, under the Project Women on Board: An Integrative approach) (PTDC/SOC-ASO/29895/2017).

**Institutional Review Board Statement:** Not applicable.

**Informed Consent Statement:** Informed consent was obtained from all subjects involved in the study.

**Data Availability Statement:** Publicly available datasets were used in this study for the analysis of descriptive representation. Data can be found here: Women and men in decision-making | Gender

Statistics Database | European Institute for Gender Equality (europa.eu). Complementary data was collected under the scope of the Women on Boards project database, available at: | Women on Boards.

**Conflicts of Interest:** The authors declare no conflict of interest.

## Appendix A

**Table A1.** Representation of women on the management and supervisory bodies of PLCs in Portugal (2018–2020).

| | 2018 | | 2019 | | 2020 | | |
|---|---|---|---|---|---|---|---|
| | PLCs (N = 39) | PLCs to Which Law No. 62 (2017) Applies (N = 14) * | PLCs (N = 38) | PLCs to Which Law No. 62 (2017) Applies (N = 23) * | PLCs (N = 38) | PLCs to Which the Minimum Threshold of 20% Applies (N = 23) * | PLCs to Which the Minimum Threshold of 33.3% Applies (N = 10) * |
| % of women on management bodies | 18% | 22% | 23% | 24% | 26% | 26% | 31% |
| % of women on supervisory bodies | 19% | 22% | 29% | 32% | 32% | 33% | 33% |
| % of women in executive positions | 9% | 12% | 13% | 15% | 14% | 15% | 11% |
| % of women in non-executive positions | 27% | 29% | 31% | 30% | 36% | 33% | 49% |
| No. of women CEOs | 1 | 0 | 1 | 1 | 1 | 1 | 0 |
| No. of women chairpersons | 1 | 0 | 1 | 1 | 2 | 2 | 0 |
| No. of women presidents of supervisory bodies | 2 | 1 | 5 | 2 | 5 | 3 | 1 |

Note: the universes presented here also include companies that do not comply with the law. * Two progressive thresholds were established for the universe of PLCs, 20% as from the first elective general meeting after 1 January 2018, and 33.3% as from the first elective general meeting after 1 January 2020 (Article 5). In this case, the thresholds apply to the total number of executive and non-executive positions. Both the thresholds and the time horizons for their compliance also apply to renewals and replacements of mandates (paragraph 5 of Article 4 and paragraph 4 of Article 5).

**Table A2.** Representation of women on the management and supervisory bodies of state-owned and local government companies in Portugal (2019–2020).

| | 2019 | | | 2020 ** | | |
|---|---|---|---|---|---|---|
| | State-Owned Companies (N = 186)* | State-Owned Companies to Which Law No. 62 (2017) Applies (N = 69) * | Local Government Companies (N = 157) ** | State-Owned Companies (N = 173) *** | State-Owned Companies to Which Law No. 62 (2017) Applies (N = 99) *** | Local Government Companies (N = 181) **** |
| % of women on management bodies | 36% | 45% | 29% | 40% | 44% | 29% |
| % of women on supervisory bodies | 43% | 47% | N/A | 42% | 45% | N/A |
| % of women in executive positions | 37% | 44% | 24% | 41% | 45% | 29% |
| % of women in non-executive positions | 30% | 38% | 35% | 38% | 43% | 37% |
| No. of women chairpersons | 29 | 18 | 23 | 31 | 25 | 25 |
| No. of women presidents of supervisory bodies | 34 | 21 | N/A | 41 | 34 | N/A |

N/A—Not applicable, as the companies for which data are available (except for one) only include the figure of the Statutory Auditor. Notes—* Nominal list of companies provided by the Office of Planning, Strategy, Evaluation and International Relations of the Ministry of Finance (GPEARI). After excluding companies in liquidation, this universe totals 186 companies. It was not possible to gather information on the composition of the boards of directors and supervisory bodies for the entire universe of state-owned companies. As a consequence, the universe under analysis corresponds to 77% and 73% in the case of the management and supervisory bodies, respectively. The information regarding the disaggregation by executive and non-executive positions corresponds to 73% of the total number of companies. ** Nominal list of companies provided by the Directorate-General of Local Authorities (DGAL). After excluding companies in liquidation, this universe totals 157 companies. It was not possible to gather information on the composition of management bodies for the entire universe of the local business sector. As a consequence, the universe under analysis corresponds to 75% in the case of the management bodies. The information regarding the disaggregation by executive and non-executive positions corresponds to 32% of the total number of companies. *** Nominal list of companies provided by the Office of Planning, Strategy, Evaluation and International Relations of the Ministry of Finance (GPEARI). After excluding companies in liquidation, this universe totals 173 companies. It was not possible to gather information on the composition of management bodies for the entire universe of state-owned companies. As a consequence, the universe under analysis corresponds to 81% and 80% in the case of the management and supervisory bodies, respectively.

The information regarding the disaggregation by executive and non-executive positions corresponds to 73% of the total number of companies. **** Nominal list of companies provided by the Directorate-General of Local Authorities (DGAL). After excluding companies in liquidation, this universe totals 181 companies. It was not possible to gather information on the composition of management bodies for the entire universe of the local business sector. As a consequence, the universe under analysis corresponds to 77% in the case of management bodies. The information regarding the disaggregation by executive and non-executive positions corresponds to 31% of the total number of companies.

**Appendix B**

**Table A3.** Descriptive statistics and mean comparison between women and men who responded to P22—Please indicate, on a scale from 1 = Completely Disagree to 5 = Completely Agree, to what extent you agree with the following statements regarding your interaction with the other members of the board of directors.

| | Mean | Standard Deviation | Minimum | Maximum | W | M | T-Test | *p*-Value |
|---|---|---|---|---|---|---|---|---|
| I always share my opinion, regardless of whether or not it is the same as that of my colleagues | 4.76 | 0.50 | 2 | 5 | 4.75 | 4.77 | 0.24 | 0.81 |
| I am always given the opportunity to share my opinion | 4.78 | 0.48 | 2 | 5 | 4.78 | 4.77 | −0.22 | 0.82 |
| My colleagues share all relevant information with me | 4.43 | 0.67 | 2 | 5 | 4.45 | 4.42 | −0.28 | 0.78 |
| I am consulted by my colleagues | 4.67 | 0.49 | 3 | 5 | 4.70 | 4.64 | −0.81 | 0.42 |
| I feel that I am treated with respect | 4.74 | 0.47 | 3 | 5 | 4.78 | 4.70 | −1.07 | 0.29 |
| I feel that my credibility is recognised | 4.72 | 0.49 | 3 | 5 | 4.73 | 4.70 | −0.48 | 0.63 |
| It is easy to get support from colleagues to share my positions and proposals | 4.48 | 0.62 | 2 | 5 | 4.49 | 4.45 | −0.40 | 0.69 |
| I have an influence on decision-making | 4.57 | 0.60 | 2 | 5 | 4.53 | 4.61 | 0.79 | 0.43 |
| Women and men who serve on the board of directors socialise outside meetings | 3.64 | 1.09 | 1 | 5 | 3.56 | 3.72 | 0.94 | 0.35 |

T—total; W—women; M—men.

**Table A4.** Descriptive statistics and mean comparison between women and men who responded to P32—Is it your experience that women and men are equally active in discussions in management and supervisory bodies?

| Is it your experience that women and men are equally active in discussions in management and supervisory bodies? | Yes, Men and Women Are Equally Active | | | No, Men Are More Active | | | No, Women Are More Active | | | Chi-Square | *p*-Value |
|---|---|---|---|---|---|---|---|---|---|---|---|
| | T | W | M | T | W | M | T | W | M | | |
| | 84.0% | 75.9% | 93.2% | 6.4% | 7.2% | 5.5% | 9.6% | 16.9% | 1.4% | 11.26 *** | 0.00 |

*** *p* < 0.01. T—total; W—women; M—men.

**Table A5.** Descriptive statistics and mean comparison between women and men who responded to P35—Considering the following topics, please indicate, on a scale from 1 = Not Important to 5 = Very Important, to what extent you consider it is important to have a balanced representation of women and men on corporate boards.

| | Mean | Standard Deviation | Minimum | Maximum | W | M | T-Test | *p*-Value |
|---|---|---|---|---|---|---|---|---|
| Equality between women and men in the company | 4.22 | 0.86 | 1 | 5 | 4.35 | 4.07 | −2.10 ** | 0.04 |

** *p* < 0.05. W—women; M—men.

**Table A6.** Descriptive statistics and mean comparison between women and men who responded to P19—Please indicate, on a scale from 1 = Not Important to 5 = Very Important, to what extent you consider equality between women and men is important.

| | Mean | Standard Deviation | Minimum | Maximum | W | M | T-Test | *p*-Value |
|---|---|---|---|---|---|---|---|---|
| For the company in general | 4.13 | 0.78 | 1 | 5 | 4.15 | 4.10 | −0.41 | 0.68 |
| For the chairperson/CEO of the corporate board you belong to | 4.18 | 0.85 | 1 | 5 | 4.18 | 4.18 | −0.01 | 0.99 |
| For the corporate board you belong to | 4.15 | 0.80 | 1 | 5 | 4.18 | 4.12 | −0.52 | 0.60 |
| For you | 4.33 | 0.82 | 1 | 5 | 4.44 | 4.21 | −1.78 | 0.08 |
| For your peers on the corporate board | 4.06 | 0.82 | 1 | 5 | 4.12 | 4.00 | −0.93 | 0.36 |

W—women; M—men.

**Table A7.** Descriptive statistics and mean comparison between women and men who responded to P20—Please indicate, on a scale from 1 = Not Important to 5 = Very Important, to what extent you consider it important that your company adopts measures to:

| | Mean | Standard Deviation | Minimum | Maximum | W | M | T-Test | *p*-Value |
|---|---|---|---|---|---|---|---|---|
| Promote a balanced participation of women and men in management and leadership positions (for example: heads of department, middle management) | 4.22 | 0.83 | 1 | 5 | 4.38 | 4.04 | −2.67 *** | 0.01 |
| Ensure equal pay for women and men on the corporate board (including variable and fixed supplements) | 4.51 | 0.92 | 1 | 5 | 4.63 | 4.39 | −1.64 | 0.10 |
| Ensure equal pay for women and men in the company (including variable and fixed supplements) | 4.52 | 0.90 | 1 | 5 | 4.62 | 4.40 | −1.54 | 0.13 |
| Promote equality between women and men in the enjoyment of rights in the field of reconciling work and family life | 4.50 | 0.81 | 1 | 5 | 4.62 | 4.38 | −1.90 | 0.06 |

*** $p < 0.01$. W—women; M—men.

**Notes**

[1]    We do not focus on symbolic representation, as its manifestations are beyond the scope of our research, which focuses on the private sector.

[2]    The *Women on Boards-PT Database* results from our study *Women on Boards: An Integrative Approach* and is based on data gathered from the following sources: financial statement reports, corporate governance reports, minutes/communications of general meetings, communications of deliberations, and company websites. Although PLCs have made relevant information about the composition of their boards available to the wider public, it was not possible to gather information on the composition of the boards for the whole universe to the public sector (state-owned and local government) companies. The limited data available refer to 2019 and 2020 and are even more restricted in relation to local government enterprises.

[3]    The interview segments were codified as follows: the first letter indicates the sex of the interviewee (W = woman; M = man), the next three letters represent the segment/group to which the interviewee's company belongs (PLC = publicly listed company; SOC = state-owned company; LGC = local government company); the following two numbers represent the interview number/chronological order.

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
