# Peer review of "Gender-Balanced Seats, Equal Power and Greater Gender Equality? Zooming into the Boardroom of Companies Bound by the Portuguese Gender Quota Law"

_socsci, doi:10.3390/socsci11100449_

Round 1
Reviewer 1 Report
Unclear sentence: "This legal additional legal obligation" - in the introduction. Please reformulate.
The citation of Vice-President Viviane Reding, the EU's Justice Commissioner could be shortened. It is not necessary to copy the whole citation. Please, just mention the concept.
The introduction should be improved including the relevance of the paper for practitioners.
It is not clear what kind of requirement the Portuguese government is asking for not before the 3rd section. Which is the mandatory gender quota? you should mention this information since from the beginning to better understand the reminder of the study. Please explain better in the introduction the specific requirements of the mandatory quota in Portugal.
"There is an important body of research demonstrating that a greater representation 169
of women on corporate boards has the potential to improve the situation of women below 170
boardroom level. " - in the theoretical framework - the paragraph starting with the sentence highlighted should be analyzed more in depth in terms of existing literature. The literature you mentioned is not very updated and just narrow. See for example:
- Magnanelli B.S., Pirolo L. (2021). Corporate Governance and Diversity in Boardrooms. Palgrave Pivot, p. 1-166, Cham, Switzerland: Palgrave Macmillan, ISBN: 9783030561192, doi: http://doi.org/10.1007/987-3-030-56120-8
which will help you also to have a better overview of the European situation in terms of mandatory gender quota, which is an aspect that should be improved in the whole paper, not only in the theoretical framework part, given that Portugal is part of the European area.
An interesting question you should answer to is the following one: when the percentage of women in the board increased, did the board size remain the same or not? This aspect is particularly relevant to understand how firms manged the change in their boardrooms. Please, develop this analysis in the results section.
It is not clear which model you used to test the significance of your results. Please, explain better the model you used.
Reinforce in the introduction why your study is relevant, which gap of the literature is filling. You stated it but the concept should be clearer since from the beginning.
In the literature review part an overview of the European situation in terms of gender quota low should be provided. This would also enhance the understanding for the reader of the relevance of your paper, as previously mentioned.
Reviewer 2 Report
Thank you for submitting your manuscript! I enjoyed reading it! Below are my comments:
- I am a corporate governance researcher so my view may be different relative to other reviewers. In my mind, I always look for the research gap that the study is trying to fulfill in the literature. In other words, the famous "so what" question needs to be addressed in the introduction section. I am not sure the author(s) succeeded with that in the introduction section. Please consider enhancing that section.
- What do you think about the differences in response rates between males and females board members (84 out of 87 females responded, and 77 out of 143 males responded)? Do you think that there is a response bias and/or non response bias? Please note that you are asking the board members to comment on a law concerning gender equality in the board room which I fully support, or course!
- How did you account for the board tenure? When were these board members elected to serve on their respective boards? Do you think that is significant given the timeline of the Portuguese governing bodies regarding the law concerning gender quota?
- I am not sure how the quantitative and qualitative data is connected. Please specify that.
- I would enhance the discussion and practical implications section along with the conclusion section.
In short, your paper addresses and important topic but is in need of further development. Best of luck with the future revisions!
Sincerely!
Author Response
Thank you for your valuable comments.
Point 1: I am a corporate governance researcher so my view may be different relative to other reviewers. In my mind, I always look for the research gap that the study is trying to fulfill in the literature. In other words, the famous "so what" question needs to be addressed in the introduction section. I am not sure the author(s) succeeded with that in the introduction section. Please consider enhancing that section.
Response: The paper adds to the Women on Boards (WoB) literature since it brings a new empirical angle by exploring the links between greater gender balance on boards, substantive equality, substantive representation and transformative change at the workplace level. We have tried to make this clearer in the introduction.
Point 2: What do you think about the differences in response rates between males and females board members (84 out of 87 females responded, and 77 out of 143 males responded)? Do you think that there is a response bias and/or non-response bias? Please note that you are asking the board members to comment on a law concerning gender equality in the board room which I fully support, of course!
Response: Thank you for your comment. Indeed, women did participate more in the survey. Ideally, we would have calculated the non-response bias. However, we do not know exactly who answered the survey, as confidentiality and anonymity were guaranteed for all respondents. Consequently, it is not possible to compare the characteristics of those who answered the survey with the characteristics of those who did not. Nevertheless, we undertook an additional analysis to compare the respondents’ positions regarding the mandated quota law. In that paper (already published in a scientific journal but not mentioned here, so as not to compromise the blind review system) about the mandated quota law in Portugal, where interviews were conducted with board members, we found that, despite the lack of a marked opposition to legally binding affirmative action measures, the current law is far from consensual. It was among men that positions and discourses were found to be more critical, while it was essentially women who tended to contradict the narrative according to which binding measures compromise meritocracy, legitimising the occupation of top management positions based on respective competencies. Thus, these findings suggest that men may have more reservations about debating the topic, which may help to explain the lower response rate.
Point 3: How did you account for the board tenure? When were these board members elected to serve on their respective boards? Do you think that is significant given the timeline of the Portuguese governing bodies regarding the law concerning gender quota?
Response: Yes, we do. Tenure was defined as the number of years since the first appointment. Given the purposes of this paper, we did not consider including either an analysis of board tenure or a comparison of the profiles of those board members appointed before and after the mandated quota law. Such an analysis has, however, been undertaken and is now available in a different paper.
Point 4: I am not sure how the quantitative and qualitative data is connected. Please specify that.
Response: Thank you for your comment. In this article, we sought to enrich the quantitative analysis with board members’ narratives through a qualitative content analysis. The interviews conducted with male and female board members proved to be a valuable and complementary source of information in addressing the intricate links between descriptive representation, substantive representation, substantive equality and transformative institutional change, in order to address the law’s potential for eliciting gender balance in the boardroom and at the workplace.
Point 5: I would enhance the discussion and practical implications section along with the conclusion section.
Response: Your point is particularly relevant. Our research team has already published a White Paper on the topic, but some of its implications can also be briefly addressed in the current paper. We have done this.
Round 2
Reviewer 2 Report
Thank you for addressing my comments!
Author Response
Dear Reviewer, The suggestions made have been incorporated - lines 297 to 308 of the manuscript.
Thank you again very much for your valuable comments.